# TEXT-TO-SQL DOMAIN ADAPTATION VIA AUTOMATED BENCHMARK TRANSFORMATION

## ABSTRACT

Text-to-SQL (Text2SQL) systems democratize access to structured databases by enabling users to retrieve data via natural language. Public Text2SQL benchmarks such as Spider and BIRD, which were introduced to measure the accuracy of Text2SQL systems, have been instrumental in driving their continual improvement. However, a Text2SQL system that performs well on a public benchmark may perform poorly when incorporated into industry applications that access proprietary databases with domain-specific vocabulary, rules and schema. While prompt-tuning or model fine-tuning techniques can significantly improve the performance of LLM-based Text2SQL systems, they require benchmarks tailored to the domain of interest. Producing a domain-specific benchmark typically requires a great deal of human expertise and labor. We introduce an automated method that generates domain-specific Text2SQL benchmarks by translating ground-truth question-SQL pairs from existing public benchmarks to a target domain. We apply our technique to the Spider and BIRD benchmarks to produce benchmarks for two target domains: asset management and patient health care. For both domains, the accuracy of various popular Text2SQL systems is typically less than half of what it is for Spider or BIRD. Fine-tuning LLM models on our generated datasets leads to substantial accuracy improvements: they can exceed a frontier model (GPT-4o) by up to 35% for asset management and up to 11% for health care. Delving into why our accuracy improvements are domain-dependent, we introduce a dataset distance metric that qualitatively correlates with the degree of improvement. In essence, our benchmark transformation technique leverages the substantial human effort expended to produce existing public benchmarks, obviating the need to repeat such effort for each domain. We plan to make our automated benchmark transformation available to the research community.

## 1 INTRODUCTION

In recent years, Text-to-SQL (Text2SQL) systems have become an increasingly popular way to access structured databases. Rather than having to master the complexities of SQL, Mongo, or other database languages, users can simply type or speak[1] their question in natural language. The Text2SQL system automatically converts the natural language to SQL, executes the SQL on one or more database tables, and returns the answer in the form of a table, a text, or a spoken response.

One key factor that has fueled the democratization of access to structured databases is the rapid rise in the ubiquity and power of Large Language Models (LLMs), which provide a far better foundation for Text2SQL solutions than the older rule-based (Popescu et al. (2004); Li & Jagadish (2014); Saha et al. (2016)) or deep-learning (Zhong et al. (2017a); Yu et al. (2018a)) technologies.

A second factor that has evolved in tandem with the first is the availability of public Text2SQL benchmarks such as WikiSQL (Zhong et al. (2017b)), Spider (Yu et al. (2018b)), BIRD (Li et al. (2023)), or Spider 2.0 (Lei et al. (2025)). Text2SQL benchmarks drive improvement in two ways. First, by enabling one to compare the accuracy of different Text2SQL solutions, benchmarks spur competition, as exemplified by leaderboards such as (https://yale-lily.github.io/spider, https://bird-bench.github.io/, https://spider2-sql.github.io/, https://openlm.ai/text2sql-leaderboard/). Second, the

---

[1] For example, see https://www.oracle.com/artificial-intelligence/speak-with-ai-about-data-using-real-time/.

benchmark datasets can be used to improve Text2SQL solutions using methods such as prompt-tuning, model fine-tuning, and agentic workflow frameworks like ReAct (Yao et al. (2023)).

Public benchmarks are typically developed through a laborious process that entails either curating or synthetically generating pairs of questions and their corresponding SQL. For example, the original Spider benchmark cost approximately 1000 person-hours spread across 11 people (Yu et al. (2018b)). Nonetheless, cross-domain public benchmarks are worth the effort because researchers and developers around the world can use them to measure and improve general Text2SQL solutions.

There is a growing set of industry applications that require Text2SQL on proprietary databases maintained by individual companies. Unfortunately, several authors (Lee et al. (2023); Tian et al. (2025); Manotas et al. (2023)) report that Text2SQL solutions that work well on public benchmarks fare poorly on new domains or applications. Techniques for improving their poor performance, such as prompt-tuning or model fine-tuning, require appropriate domain-specific benchmarks that capture the rich variety and complexity found in industry scenarios, including arcane vocabularies, obscure column names, complex rules, and tables with hundreds of columns. The problem is that, whereas the high cost of creating public benchmarks is justified by their widespread benefits, by its nature the scope of an individual domain-specific benchmark (and thus its benefit) is much smaller than that of a public benchmark, and yet the cost of developing it is the same. Therefore, automated production of Text2SQL benchmarks tailored to specific domains would be of enormous value.

In this paper, we introduce an automated method for transforming an existing benchmark into a domain-specific one. It entails extracting a templatized SQL structure from the source SQL, from which multiple target domain SQL realizations are generated algorithmically. The corresponding questions in the target domain are generated via an LLM with sufficient guardrails to ensure that the ground-truth relationship is likely to hold in the target domain. We test our method by transforming the Spider and BIRD benchmarks to two domains: asset management and patient health care. Through experiments, we a) confirm previous findings that the accuracy of Text2SQL solutions that fare well on public benchmarks drops significantly when applied to specific domains, and b) show that LLM fine-tuning based on our transformation can reverse that drop substantially in many cases.

Our main contributions include:

1. a general algorithm that reliably transforms an existing Text2SQL benchmark into a domain-specific one;

2. experiments that establish that the domain-specific benchmarks do a reasonable job of assessing the quality of Text2SQL inferencing methods applied to those domains;

3. experiments that demonstrate that the domain-specific datasets so produced support substantial improvements in Text2SQL inference accuracy via supervised fine-tuning; and

4. insights into the effectiveness of benchmark transformation based on a distance metric that can be computed on graph representations of the SQL in the source and target domains.

The remaining sections of this paper briefly review related work, describe our benchmark transformation approach, and present detailed experiments that confirm its efficacy.

## 2 RELATED WORK

Numerous prior works (Lee et al. (2023); Tian et al. (2025); Manotas et al. (2023)) have reported a significant performance decrease when LLMs, trained on public datasets such as Spider (Yu et al. (2018b)) or BIRD (Li et al. (2023)) are evaluated on new domains or applications. Standard LLM-based approaches to domain adaptation, such as model fine-tuning on specific domain data, prompt engineering, and in-context learning Dong et al. (2024) require high-quality question-SQL pairs.

Unfortunately, relatively little research attention has been devoted to addressing this fundamental data scarcity challenge, which hampers efficient Text2SQL domain adaptation. SQLsynth (Tian et al. (2025)) is one of the rare recent works that explicitly tackles this data scarcity issue by proposing a human-in-the-loop Text2SQL data annotation system. It facilitates the creation of high-quality Text2SQL datasets that can be used for fine-tuning or in-context learning via human-LLM collaboration in a structured workflow. Compared to mostly manual data creation approaches (such as

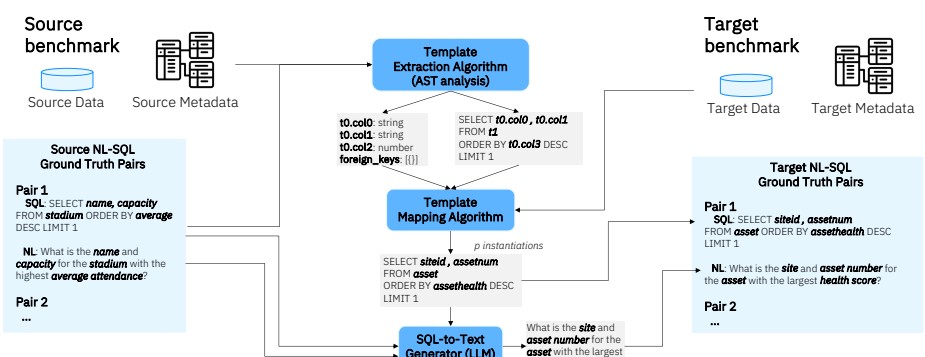

Figure 1: Overview of the benchmark transformation approach.

EHRSQL (Lee et al. (2023)), which serves as a target domain in Section 4.1), SQLsynth clearly accelerates the data annotation process. However, significant human effort is still required. In contrast, in this paper, we present a fully automated approach that generates high quality domain data without human involvement by leveraging existing publicly available Text2SQL datasets.

ScienceBenchmark (Zhang et al. (2023)) is another such domain-specific benchmark produced by a methodology that entails converting SQL to and from an Abstract Syntax Tree (AST) representation, with substitution of values in the target domain, and LLM-based generation of a question that is compatible with the generated SQL. However, their approach still requires considerable labor involving domain experts and SQL experts to create the seed question-SQL pairs from which additional pairs are generated. Since our goal is to reduce the total human labor involved, we extract our SQL templates from an existing public benchmark (Spider) rather than relying on human expertise.

OmniSQL (Li et al. (2025)) is a Text2SQL system trained on SynSQL-2.5M, a dataset synthesized with modest human effort. In contrast to our algorithmic approach, the authors use an LLM with value-masked SQL templates and various filtering techniques to compensate for the non-determinism of LLMs. Their benchmark spans a wide array of domains, serving as a large scale cross-domain benchmark similar to Spider or BIRD. However, we seek a general approach to creating datasets that support fine-tuning, prompt-tuning, or benchmarking within a specific domain.

SQL-Exchange (Daviran et al. (2025)) describes a method for transforming Text-SQL pairs across different benchmarks. A major difference from our approach is that they use an LLM to perform SQL template extraction and target domain substitution. Despite careful prompting, the authors find that the structural alignment between their source and target SQL ranges between 66% and 87% depending on details of the domains involved, and moreover the substitution error rate can be over 20%. Our algorithmic approach generates perfect structural alignment with no substitution errors. Another key difference is that SQL-Exchange is applied only between different Spider or BIRD databases, while we bridge from Spider and BIRD to real industrial target domains.

## 3 BENCHMARK TRANSFORMATION APPROACH

Consider a benchmark dataset consisting of $n$ pairs of natural language questions and corresponding ground-truth SQL statements that, when executed on the source database, produce a correct answer to the question. The objective of benchmark transformation is to convert each pair into one or more question-SQL pairs in a specified target domain such that each transformed question is relevant to the target domain and its corresponding SQL statement yields a correct answer to that question when executed on the target database.

Figure 1 overviews our benchmark transformation approach. It consists of three main steps:

1. Extract templates from ground truth SQL queries in a public dataset such as Spider;

2. Reify templates to produce new SQL queries in a different domain with a different schema;

3. For each new SQL query, generate the corresponding NL question.

Algorithm 1 outlines how the transformation algorithm is applied to each question-SQL pair in the source dataset. An Abstract Syntax Tree (AST) representation is first generated for the source SQL. Then, an abstract template is extracted from it. Next, information about the target domain schema and column values is used to randomly generate one or more substitution dictionaries that map each abstract table, column and value in the template to a compatible counterpart in the target domain. Each substitution dictionary is used to transform the source AST to a target AST. Finally, the target ASTs are converted back to target SQLs, which are parsed to ensure that they are syntactically viable and executable. A guard-railed LLM-based process produces a corresponding question for each target SQL, and the question-SQL pairs for the target domain are returned. A detailed example of the process appears in Appendix A.2. Algorithm 1 can be applied to a source-domain training dataset to produce a target-domain training dataset that supports fine-tuning or prompt-engineering an LLM model, or it can be applied to a source-domain test dataset to produce a ground-truth test set in the target domain. The remaining subsections detail the algorithm and the optional human vetting of the target dataset.

---

**Algorithm 1** Benchmark transformation

---

**Inputs**: $q_s$ (source question), $SQL_s$ (source SQL), $schema_s$ (source domain schema graph), $schema_t$ (target domain schema graph)
**Parameter**: $P$ (# desired Q-SQL pairs in target domain)
**Outputs**: $q_{tp}$, $SQL_{tp}$ for $p = 1 \ldots P$ (question-SQL pairs in target domain)

1: $ast_s \leftarrow$ **astify**($SQL_s$)
2: $template_s \leftarrow$ **extractTemplate**( $ast_s$, $schema_s$ )
3: $templateDict_s \leftarrow$ **extractDict**( $template_s$ )
4: $subDict_p \leftarrow$ **generateSubstitution**( $templateDict_s$, $schema_t$, $P$ )
5: **for** $p = 1, P$ **do**
6:    $ast_{tp} \leftarrow$ **substitute**( $ast_s$, $subDict_p$ )
7:    $SQL_{tp} \leftarrow$ **deastify**($ast_{tp}$)
8:    $q_{tp} \leftarrow$ **generateQuestion**( $q_s$, $SQL_s$, $SQL_{tp}$, $schema_t$ )
9:    **return** $q_{tp}$, $SQL_{tp}$
10: **end for**

---

## 3.1 EXTRACTING AN ABSTRACT SQL TEMPLATE

First, the standard python library *sqlglot*[2] is applied to the source SQL, $SQL_s$, to produce $ast_s$, an AST representation of $SQL_s$. Then, the function **extractTemplate** analyzes $ast_s$ in conjunction with the source domain schema $schema_s$ to extract an abstract template $template_s$, which is an AST representation in which all tables, columns and values in $ast_s$ have been replaced by generic symbols. **extractDict** places these unique generic symbols in an abstract dictionary graph $templateDict_s$ in which the nodes represent unique tables, columns and values of $SQL_s$ and edges represent the relationships among them. Node properties include a generic id and a node type (e.g. table, column or value). Column nodes have a data type property (e.g. text, integer, boolean, etc.)., while value nodes include a data type and a value property. A null data type indicates that any type is acceptable; this happens when a column appears only in a project (SELECT) statement. Edge properties include the identifiers of the two nodes for which there is an edge and the relationship type, which is either *parent* (to represent a column-table or value-column relationship) or *foreign key*. A graph representation of $templateDict_s$ can be found in Appendix A.2.

## 3.2 GENERATING A SUBSTITUTION DICTIONARY, AND SUBSTITUTION

Once $templateDict_s$ is extracted, it is input to **generateSubstitution**[3] along with the schema for the target domain $schema_t$ and a replication parameter $P$. $schema_t$ is a graph containing nodes that represent all tables and columns in the target domain and their salient properties plus edges that represent parent (table-column) or foreign key (column-column) relationships among the nodes. Among the salient node properties are a unique id, a name, and the type (table or column). Columns

---

[2]https://sqlglot.com/sqlglot.html

[3]Appendices A.1 and A.3 provide **generateSubstitution** pseudocode and a target domain schema example.

also have a data type property and a value property that specifies a set or range of representative values. While these values can be supplied manually, we prefer to extract them automatically from an existing database by executing DISTINCT commands on all columns of all tables.

Given these inputs, **generateSubstitution** attempts to randomly generate $P$ realizations of the abstract template. Each realization is expressed as a substitution dictionary $subDict_p$ that maps each node in $template_s$ to a real table, column or value in the target domain. First, the substitution dictionary is initialized such that all real tables, columns and values are set to *null*. Then, if there are any foreign key relationships among the edges of $template_s$, **generateSubstitution** loops over them first (because they are the most restrictive match). For each such foreign key edge $e_s$, **randColPair** randomly chooses from the set of foreign key relationships in $schema_t$ a pair of columns for which $(source_t, target_t)$ is compatible with $(e_s.source, e_s.target)$. In other words, we require that (1) the data types of $source_t$ and $e_s.source$ are compatible, (2) the data types of $target_t$ and $e_s.target$ are compatible, and (3) any constraints that have already been stored in $subDict_p$ from prior iterations of the loop are respected. Note that, once a mapping has been established in $subDict_p$, this defines a constraint that must be respected during subsequent iteration of any **for** loop. Once the loop over any foreign key edges is complete, the parent key edges are treated in the same way. All such edges represent either table-column or column-value relationships.

For table-column relationships, if the column $e_s.source$ has already been mapped to $source_t$, then its parent table $e_s.source$ must be mapped to the parent table of $source_t$. If $e_s.source$ has not already been mapped, then it may be selected randomly under the data type compatibility criteria above, and then its parent table mapping is completely determined as just described.

For column-value relationships, the column mapping (in this case, $subDict_p[e_s.target]$) is determined first exactly as it is for table-column relationships (i.e. it is either already determined or if not it is generated randomly). The value mapping $subDict_p[e_s.source]$) is generated randomly. There are two cases. If the value property of $source_t$ is a valueSet list, the value of $subDict_p[e_s.source]$) is selected randomly from that list. If the value property of $source_t$ is a valueRange, a value of appropriate type (e.g. number or date) is randomly generated within that range.

After looping over the foreign key and parent relationships, some column or value entries in $subDict_p$ may still be *null*. If so, a final loop is performed over any vacant entries to randomly select a target column or value from among the target tables that have already been mapped in $subDict_p$, subject to data type compatibility criteria.

A final check is performed to ensure that $subDict_p$ is not redundant with any $subDict_p$ that may have already been generated. If there is redundancy, then the **generateSubstitution** loop for $p$ is restarted at Step 4 (up to some established number of retries).

Once the substitution dictionary $subDict_p$ has been produced, each instance of each table, column, or value key $n_s$ in the $ast_s$ is replaced with its mapping $subDict_p[n_s]$ to produce the target AST $ast_{tp}$. Then, using *sqlglot*, $ast_{tp}$ is de-astified to yield the target SQL $SQL_{tp}$.

## 3.3 GENERATING THE CORRESPONDING QUESTION

A target question $q_{tp}$ corresponding to the target SQL $SQL_{tp}$ is generated by a guard-railed LLM-based procedure **generateQuestion** as follows. Using the LLM model, a prompt that includes the target $SQL_{tp}$, the source question $q_s$, the source SQL $SQL_s$ and the target domain schema graph $schema_t$ is directed to produce a target question $q_{tp}$ that captures the meaning of $SQL_{tp}$. Including the source question-SQL pair acts like a one-shot example of how to generate an appropriate question for a SQL with the same structure as $SQL_{tp}$, leading to markedly better quality of the generated target question than could be achieved without the original source pair. The candidate target-domain question-SQL pair $(q_{tp}, SQL_{tp})$ is then subjected to three LLM-based tests: syntactic correctness, SQL coverage and compatibility with question (with special emphasis on the SELECT and WHERE clauses), and SQL executability (which also includes ensuring that at least one row is produced).

If the target-domain question-SQL pair passes all three tests, it is included in the target domain dataset. Otherwise, it is either rejected or the question generator is run again to try to produce a question that will pass the tests (up to some set number of tries). The categories into which an LLM judge can classify a generated question as incorrect are the same as those used by human evaluators, as described in Appendix B.

### 3.4 HUMAN EVALUATION

When Algorithm 1 is used to generate a training set for fine-tuning or prompt-tuning, it often suffices to use that training set directly, as fine-tuning and prompt-tuning are able to tolerate a moderate number of incorrect pairs. However, if the transformation algorithm is used to create a ground truth dataset for evaluating and comparing various inferencing solutions, and if sufficient human resource is available, it may be desirable to subject the generated ground truth set to human scrutiny. For this purpose, we have implemented a UI that presents each target question-SQL pair along with other useful context such as the source question-SQL pair. The evaluator may accept, reject, or edit the pair, and may also indicate a reason for rejection. Details of the UI and the most common situations in which the evaluators chose to edit the pairs are provided in Appendix B.

## 4 EXPERIMENTS

### 4.1 SOURCE AND TARGET DOMAINS

Our experiments were conducted using two source datasets and two target domains. The first source dataset is Spider (Yu et al. (2018b)), a large-scale text-to-SQL dataset developed by 11 students that contains 200 multi-table databases covering 138 domains. It includes 10,181 question-SQL pairs that are split into 8659 train pairs and 1034 dev pairs. We apply our transformation algorithm to the Spider training pairs to produce target-domain training data in two distinct target domains: asset management and medical (patient health care). These training data are used to fine-tune LLM models, as will be explained further in Section 4.2. The second source dataset, BIRD ( Li et al. (2023)), is another large scale cross domain dataset containing 12,751 question-SQL pairs, covering 95 databases. Since we have found that experiments using BIRD as source tell much the same story as those produced using Spider, this section focuses mainly on Spider results; full details and analysis of BIRD results are included in Appendix C.

Our first target domain is asset management, which involves tracking, maintaining and optimizing physical assets like pumps, chillers, or transformers. Asset management DBs typically contain tables holding information about assets, work orders, service requests, maintenance records, etc. To create an asset management Text2SQL benchmark, we applied the benchmark transformation method described in Section 3 to the Spider train and dev datasets to generate training (**Spider.Asset.Train**) and inference (**Spider.Asset.Dev**) datasets, which contain 7000 and 832 pairs respectively. These numbers are somewhat less than the size of the Spider train and dev sets, mainly because some abstract templates could not be realized in the target domain or the generated SQL failed to produce rows. To assess the quality of **Spider.Asset.Dev**, we also conducted a human evaluation as described in Section 3.4, which produced the human-vetted ground-truth dataset (**Asset.GT**), with 725 pairs.

Our second target domain uses the EHRSQL dataset (Lee et al. (2023)), which is constructed from the MIMIC-IV clinical database (Johnson et al. (2023)) to perform text-to-SQL on electronic health records (EHR). EHRSQL includes tables like patient admissions, diagnoses, medications and so on. The EHRSQL dataset was developed with the aid of 222 hospital staff, including physicians, nurses, insurance reviewers, and health records teams. EHRSQL provides manually annotated dev and train sets, which we call **Med.Dev** (931 pairs) and **Med.Train** (5124 pairs). We generated our own training set called **Spider.Med.Train** (7000 pairs) by transforming Spider into the medical domain.

### 4.2 EXPERIMENT SETUP

For all experiments, we employ a standard execution accuracy metric (Zhong et al. (2020)). Given a Gold (ground-truth) SQL and a predicted SQL generated by the inferencing system being measured, both queries are run against the appropriate database. The reported accuracy is the percentage of predicted SQL queries that return essentially the same execution results as the gold queries.

For the fine-tuning experiments, we perform fine-tuning (Foundation-Model-Stack contributors (2024)) on the instruct version of the given model, using LoRA(Low-Rank Adaptation) (Hu et al. (2022)) on NVIDIA $A\_100$ GPUs with 80GB of memory. Each model is fine-tuned for 15 epochs using a learning rate of $1e^{-5}$. To manage memory constraints, we use a per-device batch size of 4 and gradient accumulation over 4 steps, and only fine-tune on 7b or 8b parameter instruct models. Inferencing entails prompting each LLM with standard instructions on performing text-to-SQL,

| | Spider.Dev | Asset.GT | | | Spider.Asset.Dev | |
|---|---|---|---|---|---|---|
| **Model** | base | base | ft-spider-asset | ft-bird-asset | base | ft-spider-asset |
| Granite 8b | 0.632 | 0.309 | 0.876 | 0.741 | 0.260 | 0.770 |
| Mistral 7b | 0.348 | 0.294 | 0.824 | 0.711 | 0.283 | 0.795 |
| Llama 8b | 0.7 | 0.560 | 0.860 | 0.687 | 0.493 | 0.824 |
| Mistral-M | 0.475 | 0.385 | n/a | n/a | 0.315 | n/a |
| Mistral-L | 0.695 | 0.432 | n/a | n/a | 0.363 | n/a |
| GPT-4o | 0.696 | 0.487 | n/a | n/a | 0.419 | n/a |

Table 1: **Average inference accuracy for base and fine-tuned LLM models** evaluated on Spider.Dev, GT (human-vetted ground truth) and Spider.Asset.Dev (non-vetted transformation of Spider.Dev into the asset domain).

along with the question, schema, SQL Generation rules, and formatting instructions. The LLMs then produce a predicted SQL, which is compared with the Gold SQL to compute execution accuracy.

### 4.3 RESULTS FOR ASSET MANAGEMENT DOMAIN

Experimental results in the asset management domain are summarized in Table 1. We measured the accuracies of 6 base SOTA LLMs on **Spider.Dev** (the original Spider dev dataset). The models were: granite-3.3-8b-instruct (Granite-Team, IBM (2025)), Mistral-7B-Instruct-v0.3 (Mistral-AI (2023)), Llama-3.1-8B-Instruct (Meta-AI (2024)), Mistral-Medium (Mistral-AI-Medium (2025)), Mistral-Large (Mistral-AI (2024)) and GPT-4o (OpenAI (2023)). We also measured the accuracy of fine-tuned versions of three of them[4] on two different datasets: **Asset.GT** (the human-vetted **Spider.Asset.Dev**) and **Spider.Asset.Dev** (the non-vetted transformation of Spider.Dev).

Comparing columns **Spider.Dev**/base and **Asset.GT**/base, we see that accuracy drops substantially when any base LLM model is applied to the asset management domain — a phenomenon consistent with what other authors (Lee et al. (2023); Tian et al. (2025); Manotas et al. (2023)) have reported.

The **Asset.GT**/ft-spider-asset column displays the inference accuracy obtained by fine-tuning each of the first three models on the **Spider.Asset.Train** dataset and evaluating them on **Asset.GT**. In all cases, the inference accuracy is improved by at least 30 percentage points, and is *better* than what was attained by the corresponding base model (even GPT-4o) on the original **Spider.Dev** dataset.

To assess how well the automatically-generated dataset **Spider.Asset.Dev** compares with its human-vetted counterpart **Asset.GT**, we also measured inference accuracy of each model on that dataset. Uniformly, for both base and fine-tuned models, accuracy on **Spider.Asset.Dev** tracks that of **Asset.GT**, and is always slightly lower — indicating that our auto-generated benchmark serves as a reasonable approximation (and lower bound) of the accuracy measured on a human-vetted dataset.

Column **ft-bird-asset** displays the result of fine-tuning models on **BIRD.Asset.Train** and evaluating them on **Asset.GT**. The results are qualitatively similar to those obtained using Spider as the source: fine-tuning on **BIRD.Asset.Train** produces significant improvement, albeit somewhat less than for **Spider.Asset.Train**. See Appendix C for full results and analysis of the BIRD experiments.

### 4.4 RESULTS FOR MEDICAL DOMAIN

Experimental results for the medical domain are summarized in Table 2. Comparing the accuracy of the two base columns, we see that, for all LLMs, the inference accuracy on the EHRSQL dev set plummets even more than it does in the asset management domain. Even the frontier model GPT-4o struggles, with an accuracy of only 23.7% as compared 48.7% with for asset management.

Using our framework, we transformed the Spider.train dataset to produce **Spider.Med.Train** and then used it to fine-tune the Granite, Mistral 7b, and Llama models to create their fine-tuned counterparts. The results are shown in column **Med.Dev**/ft-spider-med. There is a noticeable improvement in accuracy, with the fine-tuned Mistral 7b model out-performing GPT-4o/base by a factor of about 1.5. Nonetheless, the improvements are not nearly as impressive as they are for asset management.

---

[4]The three largest models were not fine-tuned due to infrastructure constraints.

| | Spider. Dev | Med.Dev | | |
|---|---|---|---|---|
| **Model** | base | base | ft-spider-med | ft-med |
| Granite 8b | 0.632 | 0.146 | 0.235 | 0.419 |
| Mistral 7b | 0.348 | 0.127 | 0.350 | 0.600 |
| Llama 8b | 0.700 | 0.182 | 0.182 | 0.538 |
| Mistral-M | 0.475 | 0.275 | n/a | n/a |
| Mistral-L | 0.695 | 0.316 | n/a | n/a |
| GPT-4o | 0.696 | 0.237 | n/a | n/a |

Table 2: **Inference accuracy for various base and fine-tuned LLM models** evaluated on Med.Dev. Base model performance on Spider.Dev is replicated for easy comparison.

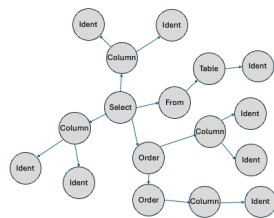

Figure 2: **Graph representation** of the SQL template *SELECT column_0, column_1, column_2 FROM table_0 ORDER BY column_0 DESC*.

We also used the human-curated EHRSQL training set (**Med.Train**) to produce another set of fine-tuned Granite, Mistral and Llama models, the results of which are shown in column **Med.FT**/ft-med. These fine-tuned models achieve higher accuracy than those fine-tuned from **Spider.Med.Train**.

When we conducted the same set of experiments using BIRD as the source benchmark, the results were qualitatively similar to those for Spider. Details may be found in Appendix C.

Across the board, models fine-tuned on **Med.Train** fared better than those fine-tuned on **Spider.Med.Train**. Why? Did the extensive human effort that produced **Med.Train** yield inherently higher-quality question-SQL pairs? A close reading of Lee et al. (2023) suggests a different possibility: the question-SQL pairs for the train and dev sets were generated by a relatively small number of templates[5]. Perhaps the abstract templates generated from Spider have greater structural diversity than those underlying both the EHRSQL datasets **Med.Dev** and **Med.Train**, and thus fine-tuning based on **Spider.Med.Train** is not as laser-focused on a narrow set of cases, resulting in lower accuracy. The next subsection delves into this hypothesis.

### 4.5 STRUCTURAL ANALYSIS OF THE SPIDER AND EHRSQL DATASETS

Our hypothesis is that a key reason why fine-tuning on **Med.Train** produces greater accuracy than fine-tuning on **Spider.Med.Train** is that **Med.Train**'s SQL structures are better aligned with those of **Med.Dev**. To test this hypothesis, we formulated two related metrics that capture the structural similarity between two SQL datasets.

To compute these metrics, we first extract templates from all SQL statements in the datasets that we wish to compare. Next, we use NetworkX (https://networkx.org/nx-guides/) to translate these templates into graphs, as illustrated in Figure 2. Then, we use the NetworkX graph edit distance measure[6] to compute a graph edit distance measure $d_{ij}$ for a given pair of graphs $i$ and $j$. Our first structural similarity metric between two datasets $I$ and $J$ is the average edit distance between all pairs of SQL template graphs in the two datasets, $\overline{d_{IJ}} = \frac{\sum_{ij} d_{ij}}{|I||J|}$, where $|I|$ and $|J|$ represent the number of SQL statements in the two datasets. Our second metric is the cumulative histogram of $d_{ij}$ as a function of edit distance, normalized to a total area of 1.

We computed $\overline{d_{IJ}}$ for I=**Spider.Med.Train** and J=**Med.Dev** to be 132.3, while that between I=**Med.Train** and J=**Med.Dev** is only 66.4. This is consistent with the much gentler rise of the cumulative frequency curves in Figure 3 for **Spider.Med.Train** (blue) than for **Med.Train** (red). Both metrics suggest that, on average, the SQL templates in **Med.Train** are much more similar to those in **Med.Dev** than to those in **Spider.Med.Train**. A similar trend occurs for asset management.

---

[5] Another factor may be that there were some non-standard assumptions about how to convert questions into SQL, i.e. generating 0 or 1 instead of "no" or "yes".

[6] See https://networkx.org/documentation/stable/reference/algorithms/generated/networkx.algorithms.similarity.graph_edit_distance.html.

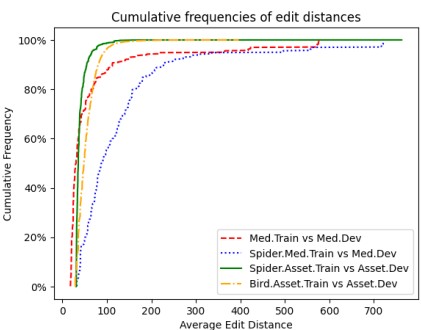

Figure 3: **Cumulative frequency of** $\overline{d_{IJ}}$ for Med.Train (red) and Spider.Med.Train (blue) vs. Med.Dev, and Spider.Asset.Train (green) and Bird.Asset.Train (orange) vs Spider.Asset.Dev.

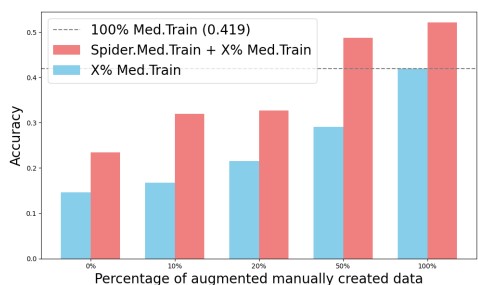

Figure 4: **Accuracy vs. % human-curated training data for fine-tuned Granite-3.3-8b.** Blue bars: X% Med.Train only, Red bars: Spider.Med.Train + X% Med.Train. Dashed line indicates accuracy achieved by Med.Train on Granite.

Recall from Table 1 that models fine-tuned on **Spider.Asset.Train** and evaluated on **Asset.GT**, produced the most outstanding results, with those fine-tuned on **BIRD.Asset.Train** being somewhat lower but still impressive. This correlates well with their respective average distances of 40.6 and 53.7, and with the relative initial behavior of the green and orange lines in Figure 3. While not thorough enough to be conclusive, these observations lend credence to the hypothesis that SQL structural similarity between the fine-tuning training and the evaluation datasets as captured by our metrics is a key determinant of the degree to which fine-tunings improve inference accuracy.

As another test, we measured the inference accuracy of Granite fine-tuned on **Med.Train** and evaluated on **Spider.Med.Dev**. When the narrowly-trained model is applied to the broader dataset, its accuracy is reduced from 0.419 to just 0.205. It seems plausible that models fine-tuned on **Spider.Med.Train** would actually outperform those fine-tuned on the hand-crafted **Med.Train** if tested on a broader set of question-SQL pairs. It would be fascinating to know whether the medical professionals who helped develop EHRSQL would value the broader range of questions that we generate.

Finally, we added various percentages of human-curated data from **Med.Train** to **Spider.Med.Train** and fine-tuned the models. Figure 4 compares the accuracy of a fine-tuned Granite model (granite-3.3 8b instruct) when trained on a given percentage of the full expensively-generated **Med.Train** with that attained by fine-tuned Granite trained on the same percentage of **Med.Train** plus the zero-cost **Spider.Med.Train**. In all cases, supplementing **Med.Train** with **Spider.Med.Train** results in significant accuracy improvements. At 50%, our fine-tuned model's accuracy exceeds that attained using 100% **Med.Train**, and at 100% it reaches 0.521 — 10.2% better than **Med.Train** only.

## 5 CONCLUSION

Text2SQL solutions that work well on public datasets often fare poorly when applied in domain-specific applications, as our experiments have corroborated. For a Text2SQL application to be truly successful in a given industry domain, it critically needs a benchmark that is tailored to it. To date, the cost of creating such benchmarks has been prohibitive. Our automated benchmark transformation method, which we plan to release to the research community, solves this problem by leveraging the substantial human effort that has already been poured into common public benchmarks such as Spider and BIRD, obviating the need to repeat such effort for each domain. As we established experimentally, fine-tuning medium-sized LLM models on transformed Spider and BIRD datasets produced up to a 35% accuracy improvement over base models for the asset management domain and up to 11% for the patient health care domain. This renders them more accurate than GPT-4o, a frontier model. Moreover, we can leverage our transformed datasets to obtain reasonably good measures of the inference accuracy of Text2SQL solutions in target domains. Finally, we introduced dataset distance metrics that correlate well with the efficacy of fine-tuning in a given domain, warranting further investigation of this hypothesis in future work.

## 6 REPRODUCIBILITY

The benchmark transformation approach described in this paper is documented in detail in Section 3 in the form of text and pseudo-code. The appendix provides further details of the transformation process, including pseudo-code for the main subroutine **generateSubstitution** in Appendix A.1, a detailed example of the overall transformation process in Appendix A.2, and details of the target domain schema in Appendix A.3. Readers interested in the optional step of human vetting will find a writeup about the human evaluation UI in section 3.4 and Appendix B.

In section 4.1, we provide references to the Spider and BIRD source datasets that would enable an interested reader to download them. We provide links to the previously-published patient health care dataset in 4.1, so this should be replicable. The asset management dataset is proprietary and therefore we are not free to share it, but we hope that interested readers will try our methods on other databases and we believe that our detailed description of the transformation process will enable them to do so. Section 4.2 details the LLM models we used and the techniques, hardware and parameters we employed for fine-tuning them. We clearly identify the graph isomorphism software and techniques that we used for the structural dataset analysis in Section 4.5.

We provide all of our source code in the supplementary materials. These materials also contain the prompts used for LLMs responsible for generating questions from SQL or judging the compatibility of the generated text/SQL pairs. If the paper is accepted, we plan to make the transformed datasets available to the research community.

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

# A   Appendix: Methodology details

This appendix describes in greater detail several entities mentioned in the Methodology section of the paper (section 3): the **generateSubstitutionAlgorithm**, abstract SQL template and dictionary, and the target domain schema format.

## A.1   generateSubstitution Algorithm

Here is pseudocode for the algorithm **generateSubstitution** that is described in section 3.2.

---

**Algorithm 2** generateSubstitution

---

**Inputs**: $template_s$ (extracted template graph), $schema_t$ (target domain schema graph), $P$ (# desired realizations)

**Outputs**: $subDict_p$ (substitution dictionary)

1: $fk \leftarrow$ set of foreign key edges in $template_s.edges$
2: $par \leftarrow$ set of parent edges in $template_s.edges$
3: **for** $p = 1, P$ **do**
4:     **for each** $node \in template_s.nodes$ **do**
5:         $subDict_p[node] \leftarrow null$
6:     **end for**
7:     **for each** $e_s \in fk$ **do**
8:         $(source_t, target_t) \leftarrow$ **randColPair**$(subDict_p, schema_t)$
9:         $subDict_p[e_s.source] \leftarrow source_t$
10:         $subDict_p[e_s.target] \leftarrow target_t$
11:     **end for**
12:     **for each** $e_s \in par$ **do**
13:         $(source_t, target_t) \leftarrow$ **randColPair**$(subDict_p, schema_t)$
14:         $subDict_p[e_s.source] \leftarrow source_t$
15:         $subDict_p[e_s.target] \leftarrow target_t$
16:     **end for**
17:     **for each** $n_s \in subDict_p$ **do**
18:         **if** $subDict_p[n_s] == null$ **then**
19:             $subDict_p[n_s] \leftarrow$ **randCol**$(subDict_p, schema_t)$
20:         **end if**
21:     **end for**
22:     **return** $subDict_p$
23: **end for**

---

## A.2   TRANSFORMATION EXAMPLE

This section presents an example of several stages of the transformation process that maps a given text/SQL pair in the Spider domain to one in the asset management domain.

In this example, the original Spider question $q_s$ is:

> "What is the total number of unique official languages spoken in the countries that are founded before 1930?".

and its associated ground-truth SQL $SQL_s$ is given as:

```
SELECT count(DISTINCT T2.Language)
FROM country AS T1
JOIN countrylanguage AS T2 ON T1.Code = T2.CountryCode
WHERE IndepYear < 1930 AND T2.IsOfficial = "T"
```

First, *sqlglot* processes $SQL_s$ to produce the abstract template $ast_s$, a verbose AST expression that is too long to list here. *sqlglot* also produces a unique set of tables, columns and values appearing in $SQL_s$, which are replaced by generic symbols in $ast_s$ to form $template_s$. Next, **extractDict** extracts from $template_s$ the template dictionary $templateDict_s$, depicted as a graph in Figure 5.

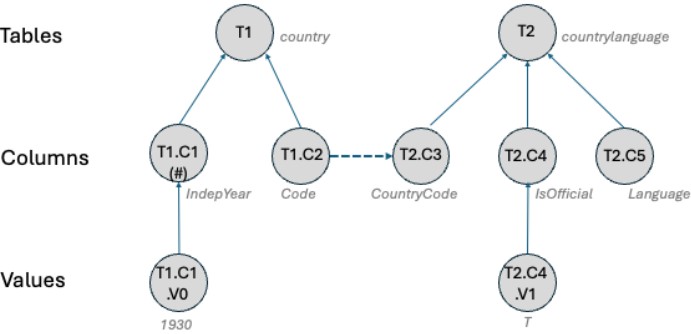

Figure 5: **Template dictionary example.** Template dictionary extracted automatically from $SQL_s$, represented as a graph containing 2 table nodes, 5 column nodes, and 2 value nodes. Symbolic names appear inside the nodes. For clarity, the original names from the source (Spider) domain are included in italics just outside the nodes; this information is not retained in the template dictionary itself. Parent relationships between column and table or value and column are represented as solid edges; foreign key relationships between columns are represented as dashed arrows. The only column with a dataType constraint is T1.C1, derived from the IndepYear column of the country table. Its dataType must be Number because it appears in a WHERE clause with a numeric operator (less than); T2.C4 is not similarly constrained because the equality operator in the WHERE clause involving the IsOfficial column does not constrain the dataType. The dataTypes of the other columns are not constrained either because they only appear in SELECT and JOIN statements.

The algorithm **generateSubstitution** operates on $templateDict_s$ to produce a set of $P$ substitution dictionaries, $subDict_p$, one of which is depicted in Figure 6.

This substitution dictionary is used to generate $ast_{tp}$, which is de-astified to produce the following transformed SQL $SQL_{tp}$:

```
SELECT COUNT(DISTINCT workorder.commodity)
FROM assetmeter
JOIN workorder ON assetmeter.assetnum = workorder.assetnum
WHERE assetmeter.assetmeterid < 270 AND workorder.orgid = 'HOUSDC'
```

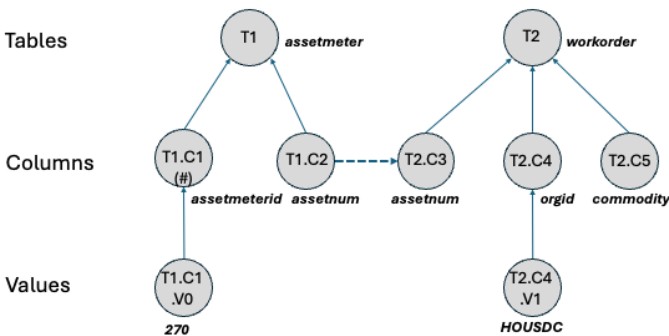

Figure 6: **Substitution dictionary example.** Substitution dictionary produced by **generateSubstitution**. The structure is the same as in Figure 6, but the nodes are now annotated with the substituted values from the target domain (asset management).

Next, **generateQuestion** is applied to this transformed SQL plus the original $q_s$ and $SQL_s$ given above to produce the corresponding $q_{tp}$:

> "How many distinct commodities are involved in work orders for assets with a
> meter ID less than 270 and belonging to the organization HOUSDC?"

Finally, the pair $(q_{tp}, SQL_{tp})$ is returned as a transformation of the original Spider question/SQL pair to the asset management domain.

## A.3 TARGET DOMAIN SCHEMA EXAMPLE

This section of the appendix exemplifies the target domain schema $schema_t$ by showing snippets of it from the medical domain.

The target domain schema may be expressed as a graph with nodes and edges. Just as is the case for the template, the nodes represent tables or columns and their properties, while the edges represent either parent (table-column) or foreign key (column-column) relationships. All nodes have a unique id, a name, and a type (table or column). A typical table node is:

```
{
    "id": "admissions",
    "name": "admissions",
    "type": "table"
}
```

Column nodes have two additional properties: a data type and a property ValueSet that captures the set of values that the column may take. For columns of type text, we typically limit the set of allowed values to no more than 20 or so. A typical example is:

```
{
  "id": "admissions.marital_status",
  "name": "marital_status",
  "type": "column",
  "dataType": "text",
  "valueSet": [
    "single",
    "married",
    "widowed",
```

```
756        "divorced"
757      ]
758  }
759
```

For columns to boolean type, the possible values are assumed to be true or false, while for numbers and dates the minimum and maximum values for each column are identified and stored as a ValueRange property of the node. Here is an example of a node definition for a numeric column:

```
764  {
765    "id": "admissions.age",
766    "name": "age",
767    "type": "column",
768    "dataType": "number",
769    "valueRange": [21, 91]
770  }
```

The edges of the target domain schema graph have the same format as the abstract template graph described in Section 3.1, e.g.

```
774  {
775    "source": "admissions.subject_id",
776    "target": "patients.subject_id",
777    "type": "foreignKey"
778  }
```

## B  APPENDIX: HUMAN EVALUATION

We implemented a UI that presents each target question-SQL pair to the evaluator. It also provides the source question-SQL pair for context, as well as definitions of any target-domain tables or columns referred to in the target-domain SQL. The evaluator can indicate whether they wish to accept or reject the pair by clicking on the button. The evaluator can indicate a reason for rejecting the pair; the rejection categories are:

- *missing_column*. The NL (natural language) question does not include a column name that is mentioned in the SQL query, either in the SELECT or WHERE clause.

- *missing_table*. The NL (natural language) question does not mention the table that is being referenced in the SQL, either directly or indirectly.

- *missing_constraint*. There is a constraint, like "show only one", or "order by" that is present in the SQL but not mentioned in the NL question.

- *missing_condition*. There is a condition (e.g. greater than, less than, etc.) that appears in the SQL but is not mentioned in the NL question.

- *other*. Any other reason why the NL question is an incorrect representation of the SQL.

Users may also choose to edit the SQL and/or the question to make it more appropriate to the domain. The most common situations in which evaluators have chosen to edit the target-domain question-SQL pair include:

- Correcting pluralization in the question, e.g. from "Which asset..." to "List assets that...". Often, incorrect pluralization stems from a flaw in the source question that implicitly assumes that a single answer should be returned. This happens relatively frequently for the Spider domain. One possible explanation is that the Spider tables tend to be quite small, often with just a few dozen rows, and often the result set for a query contains just one row. This may have influenced the Spider authors to write questions that anticipate a singular answer. It is only when the question-SQL pair is translated to another domain that this assumption is exposed as incorrect. Fortunately, incorrect pluralization is mainly a cosmetic issue, as it tends not to interfere with inferencing.

- Modifying a column to a more appropriate one of the same type. For example, a question asking for the average or sum of IDs is technically legitimate and thus not harmful to the data set, but it is rather unnatural. Additional automation to address this issue would be a good idea, and relatively simple to implement.

## C  APPENDIX: ADDITIONAL BIRD EXPERIMENTS

### C.1  ASSET DOMAIN

Table 3 summarizes experimental results in which the source domain was BIRD and the target domain was asset management. The first column shows the accuracy of the SOTA LLMs on the BIRD dev set. We measured the accuracy of the fine-tuned versions of three of them, Granite-3.3-8b-instruct, Mistral-7b-Instruct-v0.3 and Llama-3.1-8b-Instruct, on two different datasets: **Bird.Asset.Dev** (the non-vetted transformation of Bird.Dev to the asset domain), and **Asset.GT** (the human-vetted **Spider.Asset.Dev**).

BIRD is designed as a harder cross-domain benchmark than Spider. The added complexity arises from extra information in the form of evidence that is required to identify the right aggregations and relationships between the tables and columns, in addition to the natural language question. This complexity reduces the accuracy of the SOTA methods on the BIRD benchmark when compared to Spider. As can be seen by comparing the columns **Bird.Dev/**base and **Bird.Asset.Dev/**base, the accuracy of SOTA models drops significantly when evaluated on **Bird.Asset.Dev** instead of the original benchmark **Bird.Dev**, which is consistent with what we observed for Spider in Section 4.3.

When the Granite, Mistral and Llama models are fine-tuned with **Bird.Asset.Train** (the dataset derived by transforming **Bird.Train** to the asset domain), the accuracy increases substantially, as was the case for Spider. Comparing columns **Bird.Asset.Dev/**ft-bird-asset and **Bird.Asset.Dev/**base, the accuracy increases by more than a factor of 3 for Mistral-7b, nearly 2 for Granite-8b, and about 1.5 for Llama-8b. The absolute accuracies are not as high as for Spider (generally about 0.6 compared with about 0.8), but once again they are even higher than for the base models evaluated in the source domain (**Bird.Dev/**base). As was also the case for the Spider experiments reported in Table 1, the evaluated accuracies of the fine-tuned models are somewhat higher on the hand-curated **Asset.GT** than they are on **Bird.Asset.Dev**. We attribute this to two factors. First, as was true for Spider, the hand-curated dataset is cleaner and thus a truer assessment of the accuracy of the fine-tuned model. Second, the alignment between **Bird.Asset.Train** and **Asset.GT** is not as strong as it is between **Spider.Asset.Train** and Asset.GT, as **Asset.GT** was derived from Spider.

| | Bird.Dev | Asset.GT | | Bird.Asset.Dev | |
|---|---|---|---|---|---|
| **Model** | base | base | ft-bird-asset | base | ft-bird-asset |
| Granite 8b | 0.411 | 0.309 | 0.741 | 0.321 | 0.633 |
| Mistral 7b | 0.310 | 0.294 | 0.711 | 0.205 | 0.637 |
| Llama 8b | 0.462 | 0.560 | 0.687 | 0.392 | 0.587 |
| Mistral-M | 0.293 | 0.385 | n/a | 0.201 | n/a |
| Mistral-L | 0.375 | 0.432 | n/a | 0.327 | n/a |
| GPT-4o | 0.512 | 0.487 | n/a | 0.271 | n/a |

Table 3: **Average inference accuracy for base and fine-tuned LLM models** evaluated on Bird.Dev, GT (human-vetted ground truth) and Bird.Asset.Dev (non-vetted transformation of Bird.Dev into the asset domain).

These findings are consistent with the structural analysis reported in Section 4.5. Recall from Figure 3 that the cumulative frequency for **Bird.Asset.Train** vs **Spider.Asset.Dev** (the yellow line) rises almost as quickly in the beginning as for **Spider.Asset.Train** vs **Spider.Asset.Dev** (the green line), indicating that most graph edit distances are very small, but not quite so small as for Spider. This correlates with the accuracies for models fine-tuned on BIRD being somewhat less than for Spider, but still reasonably good — indicating that there is at least moderate similarity between the SQL templates of Spider and BIRD.

| | Bird.Dev | Medical.Dev | | |
|---|---|---|---|---|
| **Model** | base | base | ft-bird-med | ft-med |
| Granite 8b | 0.411 | 0.146 | 0.181 | 0.419 |
| Mistral 7b | 0.310 | 0.127 | 0.185 | 0.600 |
| Llama 8b | 0.462 | 0.182 | 0.175 | 0.538 |
| Mistral-M | 0.293 | 0.275 | n/a | n/a |
| Mistral-L | 0.375 | 0.316 | n/a | n/a |
| GPT-4o | 0.512 | 0.237 | n/a | n/a |

Table 4: **Inference accuracy for various base and fine-tuned LLM models** evaluated on Med.Dev. Base model performance on Bird.Dev is replicated for easy comparison.

## C.2  MEDICAL DOMAIN

Table 4 summarizes the results of SOTA models on Bird.Dev and Med.Dev, the effect of fine-tuning using **Med.Train** (human-generated domain-specific data from EHRSQL), and fine-tuning using **Bird.Med.Train** (ft-bird-med), which was generated by transforming the data from **Bird.Train** into the medical domain. The fine-tuning was performed on the usual 3 models: Granite 8b, Mistral 7b and Llama 8b. As was the case for Spider, the base values of these models in the medical domain are quite low when compared to the base values in the asset domain, which we attribute to the very specific nature of the questions and SQL patterns in **Med.Dev**. The fine-tuning of the instruct models is not as successful using **Bird.Med.Train** as it was using the **Spider.Med.Train**, due to the inherent differences in templates between the Bird.Train data and the Med data. Additionally, since the SOTA models already have a much lower accuracy on **Bird.Dev**, some of that loss in accuracy also affects the domain conversion process into the medical domain, which has some assumptions needed in order to map the natural language question into SQL.

The structural analysis hypothesis of Section 4.5 also holds for BIRD applied to the medical domain. The average graph edit distance is 166.1 for **Bird.Med.Train** vs **Med.Dev**, compared to the average distance of 66.4 for **Med.Train** vs **Med.Dev**, which is consistent with the much lower accuracies of ft-bird-med relative to ft-med.