# OpenReview forum: "Text-to-SQL domain adaptation via automated benchmark transformation"
_ICLR.cc/2026/Conference — Submitted to ICLR 2026_

### Official Review · Reviewer_iADb · 2025-10-26

**Soundness:** 2
**Presentation:** 2
**Contribution:** 3
**Rating:** 2
**Confidence:** 5

**Summary:**

This paper presents an automated method for generating domain-specific Text-to-SQL benchmarks by transforming existing public datasets (e.g., Spider, BIRD) into new target domains. The method extracts an Abstract Syntax Tree (AST) from each SQL query, builds a schema-agnostic template, substitutes tables and columns using the target domain schema, and finally uses an LLM-based generator to produce corresponding natural-language questions.

The goal is to alleviate the high cost of human-curated Text-to-SQL datasets and enable domain adaptation for LLM-based Text-to-SQL models. The authors evaluate the approach by transforming Spider and BIRD into two target domains (asset management and health care) and fine-tuning several models (Granite-8B, Mistral-7B, Llama-8B). Reported results show accuracy gains of up to 35% for asset management and 11% for health care, compared to base models.

**Strengths:**

* **Motivated problem**: Addresses the lack of domain-specific text-to-SQL datasets for fine-tuning or evaluation.

* **The framework**: The idea of transforming existing benchmarks via AST-based schema substitution and LLM-based question regeneration is well-grounded.

* **Empirical evaluation**: Experiments with multiple models and domains provide an initial validation of feasibility, showing measurable gains in downstream performance.

* **Potential impact**: If robust and reproducible, the approach could significantly reduce the cost of domain adaptation for industrial Text-to-SQL applications.

**Weaknesses:**

* **W1. Incomplete methodological details**.
  The paper describes the algorithm in a procedural, code-oriented manner without sufficient abstraction or formalization. For instance, both source and target schemas are represented as ASTs, but it remains unclear how structural mismatches are resolved:

  * What happens when a target schema lacks a compatible column, key, or relationship?

  * How does the substitution handle differing schema cardinalities or missing join paths?

  Without this clarification, the claim of “perfect structural alignment” is not well supported.

* **W2. Limited conceptual framing**.
The presentation focuses heavily on low-level function calls and pseudocode, which obscures higher-level insights. A clearer articulation of research questions (e.g., how much semantic fidelity is preserved? how do LLMs affect question quality?) would make the contribution more scientific and less engineering-oriented.

* **W3. Data availability and verification issues**.
The asset management dataset, which yields the largest reported improvement (35%), is not included in the submission or publicly available. This prevents independent verification and weakens the empirical claims. Clarifying whether the dataset or schema will be released is essential for reproducibility.

* **W4. Overstated comparison to related work**.
The authors contrast their method with prior approaches such as SQL-Exchange, claiming “perfect structural alignment.” However, this assertion is not rigorously evaluated or quantified. A formal comparison (e.g., alignment accuracy, substitution success rate) would strengthen this claim.

* **W5. Limited analysis of quality and generality**.
While accuracy gains are reported, the paper does not analyze the semantic validity of generated question–SQL pairs or whether improvements transfer beyond the tested domains. Human evaluation is mentioned but not detailed quantitatively.

**Questions:**

1. How does the substitution algorithm behave when the target schema has no foreign key or column corresponding to the source? Are partial matches or fallbacks allowed?

2. Could you clarify why the performance results in Table 1 omit some models (Mistral-M, Mistral-L, GPT-4o)?

3. Will the asset management schema/data be made public? It does not seem to be public and is not included in the supplementary materials.

4. How are incorrect or semantically invalid question–SQL pairs filtered during generation? How are "LLM-based tests" performed?

---

### Official Review · Reviewer_EAaa · 2025-10-28

**Soundness:** 3
**Presentation:** 3
**Contribution:** 2
**Rating:** 6
**Confidence:** 4

**Summary:**

This paper proposes a method for automatically transforming benchmark data to specific domains, aiming to avoid the high cost of manually creating benchmarks for each new domain. The method achieves robust target-domain SQL generation through an algorithm based on Abstract Syntax Trees (ASTs) and dictionary graphs, and utilizes the few-shot capabilities of LLMs to generate corresponding target-domain questions. In downstream healthcare and asset management domains, models fine-tuned on training data generated by this method (from Spider or BIRD) show significant gains compared to base models. This indicates that the method robustly achieves high-quality domain data generation while successfully avoiding expensive manual annotation overhead.

**Strengths:**

1. This paper aims to solve the problem of domain-specific data scarcity and proposes an automated solution. It avoids high manual overhead and demonstrates the high quality of the generated data in experiments on two downstream tasks, thus possessing high practical value.
2. The purely algorithmic approach to target-domain SQL generation ensures a high correctness rate for the SQL. The use of test cases to verify LLM-based question generation enhances, to some extent, the correctness of the questions and their alignment with the SQL, ensuring high data quality overall.
3. Experiments in the downstream healthcare and asset management domains prove that training on the target-domain data generated by this method (from Spider or BIRD) leads to significant performance improvements.

**Weaknesses:**

1.As the authors mentioned in the paper, the performance in the downstream domain depends on the coverage of the SQL templates in the source benchmark data.

2.It is suggested to add the performance of a model trained directly on the Spider training data as a baseline in the experiments in Table 1, in order to strengthen the validation of the template-based data injection method.

**Questions:**

1. The `generateSubstitution` algorithm prioritizes matching foreign key constraints to ensure the template-substituted SQL maintains valid relational semantics. For downstream tasks involving denormalized wide-table structures or schemas where foreign key constraints have been removed, can the current substitution algorithm still guarantee the validity of these relational semantics?
2. When using an LLM to generate natural language questions, even with the "LLM-generated Test" filtering mechanism, a certain proportion of semantic inconsistencies between the natural language and the SQL may still exist. What is the approximate proportion of such semantic inconsistencies (noise) in the generated training data? Does this noisy data impact the performance on downstream tasks? Is manual filtering necessary?
3. For downstream tasks where the existing benchmark's template coverage is low, are there any measures to mitigate the performance loss from the transfer?

---

### Official Review · Reviewer_CfDz · 2025-10-29

**Soundness:** 2
**Presentation:** 3
**Contribution:** 2
**Rating:** 4
**Confidence:** 3

**Summary:**

The paper presents a method that adapts an existing Text-to-SQL benchmark and its queries to a different domain through automated transformation steps. The motivation lies in the scarcity of domaion-specific training data and in previous observations that Text-to-SQL systems that perform well on public benchmarks such as Spider or BIRD, perform poorly on unseen, domain specific databases. The method in the paper leverages AST extraction to create templates from existing benchmark queries and then fills in these templates with values from the new domain (table, column, values). Then a LLM is used to create the corresponding natural language question to the newly generated SQL query. The data generation method is evaluated for two domains (1) a syntactically created asset management domain and a health domain (based on the EHRSQL dataset). The authors create transformations from both Spider and BIRD. In both cases, they show that base models perform worse than on the source dataset and that fine-tuning improves Text-to-SQL performance.

**Strengths:**

S1: The motivation of the paper is sound. Creating domain-specific training data for Text-to-SQL is challenging, but usually improves results via fine-tuning or in-context-learning.

S2: The results of the paper show clearly that a dataset derived from a popular and public benchmark dataset will have a lower execution accuracy and that fine-tuning with the transformed data will boost performance.

S3: The analysis and thoughts on the structural alignment between train and test dataset for the EHRSQL dataset are interesting.

**Weaknesses:**

W1: It is not surprising that models will perform worse on newly created datasets compared to well-known benchmarks such as Spider, which surely models have been trained on. The same goes for the fact that then fine-tuning with the generated data will improve results.

W2: The initial goal of the paper is to improve Text-to-SQL performance for specific domains and databases. For example for the internal database of a company. The experiments are however not testing that, but are testing again on the generated Text-Query pairs. It would be more interesting to show how the method can improve performance of real user queries based on the syntactic data generation approach. For example on the dataset in [1]. The results on the medical dataset shows already that the transformed training data achieves a smaller performance gain on the original dev dataset (which is template based) than on the from Spider derived dev data.

W3: There are other methods for syntactic data generation, for example Omnisql, SQLsynth (both mentioned in the paper), SyntaxSQLNet [2], SQLord [3], SQLForge [4], CodeS [5], SENSE [6]... Why did the authors not implement any of these methods and compare the capability of their syntactic data generation against the authors method?

W4: Human evaluation. I miss some details on the results of the human evaluation. How many errors are in the created Text-to-SQL pairs? In which categories?

[1] Fürst, Jonathan, et al. "Evaluating the Data Model Robustness of Text-to-SQL Systems Based on Real User Queries." EDBT. 2025.

[2] Yu, Tao, et al. "Syntaxsqlnet: Syntax tree networks for complex and cross-domaintext-to-sql task." arXiv preprint arXiv:1810.05237 (2018).

[3]Cheng, Song, et al. "SQLord: A Robust Enterprise Text-to-SQL Solution via Reverse Data Generation and Workflow Decomposition." Companion Proceedings of the ACM on Web Conference 2025. 2025.

[4] Guo, Yu, et al. "SQLForge: Synthesizing Reliable and Diverse Data to Enhance Text-to-SQL Reasoning in LLMs." arXiv preprint arXiv:2505.13725 (2025).

[5] Li, Haoyang, et al. "Codes: Towards building open-source language models for text-to-sql." Proceedings of the ACM on Management of Data 2.3 (2024): 1-28.

[6] Yang, Jiaxi, et al. "Synthesizing text-to-SQL data from weak and strong LLMs." arXiv preprint arXiv:2408.03256 (2024).

**Questions:**

See also weaknesses above.

Q1: How many errors are in the generated data?
Q2: Can you show performance gains with your method by creating queries from a dataset like Spider for a domain-specific dataset consisting of real user queries such as FootballDB [1]?
Q3: Can you then compare your data generation method to other methods of syntactic data generation and compare in terms of the needed efforts as well as

---

### Official Review · Reviewer_1Pvp · 2025-11-01

**Soundness:** 3
**Presentation:** 3
**Contribution:** 2
**Rating:** 2
**Confidence:** 2

**Summary:**

This paper introduces an automated method for transforming public Text-to-SQL benchmarks into domain-specific ones by extracting SQL templates, substituting domain-relevant elements (e.g., tables, columns, values), and generating corresponding natural language questions via LLMs with guardrails. The approach is applied to two target domains: asset management (proprietary) and patient health care (based on EHRSQL). Experiments show that state-of-the-art LLMs (e.g., GPT-4o, Mistral, Llama) achieve high accuracy on source benchmarks but drop significantly (often to <50%) on domain-specific data. Fine-tuning medium-sized models (7B-8B parameters) on the transformed datasets yields substantial improvements: up to 35% absolute gain in asset management (surpassing GPT-4o) and up to 11% in health care. The authors also propose a graph-based dataset distance metric (using NetworkX graph edit distance) that correlates with fine-tuning efficacy, explaining domain-dependent results (e.g., better alignment in asset management). Contributions include the algorithm, experimental validation, fine-tuning improvements, and insights into dataset similarity. The method leverages existing benchmarks to reduce human effort for domain adaptation, with plans to release code and datasets.

**Strengths:**

- The paper tackles the well-known gap between cross-domain public benchmarks and industry databases with domain-specific schemas and vocabularies.
- The automated benchmark transformation is a simple yet effective algorithmic approach that addresses a real pain point in Text2SQL domain adaptation.
- Fine-tuning on the transformed training sets gives notable gains in asset management and medical setting.

**Weaknesses:**

- Asset management data and schema details are not shareable, which limits external validation and leaderboard comparability.
- Results are strong in asset management, but improvements in EHRSQL are modest. More evidence across diverse domains (beyond the single publicly released domain) would strengthen the validity and the impact of the claim.
- Many of the models appear outdated. For example, Qwen3-32B may outperform the models reported in the paper without fine-tuning.
- Now, the machine learning literature offers many fine-tuning methods (e.g., DPO and GRPO), yet this work lacks novelty by focusing solely on data augmentation and supervised fine-tuning without exploring these alternatives.

**Questions:**

See above

---

### Meta-Review · Area_Chair_pbrx · 2025-12-20

**Summary:**

This paper is about a system for generating domain-specific Text2SQL question-answer datasets that could be used to benchmark a Text2SQL system on the specific domain where it would be applied.

Reviewers brought up several concerns which I would expect to be addressed before acceptance:
- Proprietary datasets were used for some of the results with no indication that they would eventually be released, which harms the reproducibility of the work.
- Improvements on the public datasets were modest compared to the proprietary ones without explanation of why
- Human evaluation was mentioned but not detailed in a sufficient way
- Lacking evaluation of noise or errors that are generated by the system
- Some claims are made about perfect structural alignment that are not justified

**Reviewer Concerns:**

The authors did not provide a rebuttal, so all of the above concerns remain outstanding. Since the peer review of this work was not completed due to the authors' lack of engagement, I must firmly recommend rejection.

**Reviewer Scores:**

Since no rebuttal was made, I assume reviewers have not changed their opinions.

Reviewer 1Pvp - 2 -> 2

Reviewer CfDz - 4 -> 4

Reviewer EAaa - 6 -> 6

Reviewer iADb - 2 -> 2

---

### Decision · Program_Chairs · 2026-01-26

Reject